# Neural Field Dynamics Model for Granular Object Piles Manipulation

**Shangjie Xue,** **Shuo Cheng, Pujith Kachana, Danfei Xu**
Georgia Institute of Technology
Email:{xsj,shuocheng,pkachana3,danfei}@gatech.edu

**Abstract:** We present a learning-based dynamics model for granular material manipulation. Drawing inspiration from computer graphics' Eulerian approach, our method adopts a fully convolutional neural network that operates on a density field-based representation of object piles, allowing it to exploit the spatial locality of inter-object interactions through the convolution operations. This approach greatly improves the learning and computation efficiency compared to existing latent or particle-based methods and sidesteps the need for state estimation, making it directly applicable to real-world settings. Furthermore, our differentiable action rendering module makes the model fully differentiable and can be directly integrated with a gradient-based algorithm for curvilinear trajectory optimization. We evaluate our model with a wide array of piles manipulation tasks both in simulation and real-world experiments and demonstrate that it significantly exceeds existing methods in both accuracy and computation efficiency. More details can be found at https://sites.google.com/view/nfd-corl23/.

**Keywords:** Deformable Object Manipulation, Manipulation Planning

## 1 Introduction

Granular objects such as beans, nuts, and ball bearings are ubiquitous in daily life and industry [1], making accurate granular modeling and manipulation essential for various robotic tasks. However, such modeling is challenging due to the large number of particles and complex dynamics, as well as the properties of granular materials such as size, shape, friction, and contact mechanics. Overcoming these challenges is crucial to developing accurate and efficient learning-based models that enable robots to perform tasks involving granular materials.

To address these challenges, recent efforts have been made to model granular pile dynamics using deep latent dynamics models directly from pixel input [2]. However, such models have shown limitations in capturing the dynamics, under-performing a linear dynamics model due to a lack of inductive biases [2]. Alternatively, physics-inspired concepts such as particles lend themselves as strong inductive biases for deep dynamics models. A long line of works approximate a system with a collection of particles and model inter-particle dynamics [3, 4]. How-

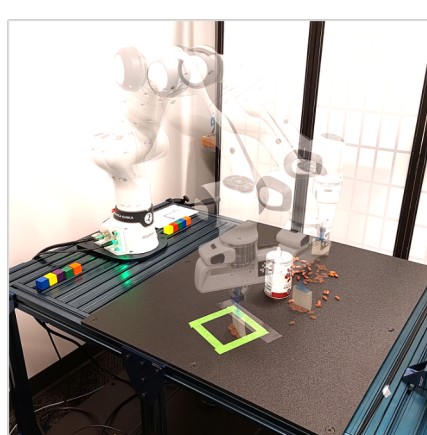

Figure 1: We present a flexible learning-based dynamics model that allows robot to manipulate object piles into target configurations while avoiding obstacles.

ever, while conventional particle-based techniques have demonstrated impressive accuracy, their memory and computational costs grow superlinearly with the number of particles [5, 6], posing scalability challenges for their application to granular material manipulations. Moreover, these methods assume that the underlying particles can be tracked, limiting their real-world applicability.

7th Conference on Robot Learning (CoRL 2023), Atlanta, USA.

In this work, we argue that *field-based* representations are better-suited for modeling granular object piles. By representing the space in which the physical system resides as a density field with discrete sampling positions, we can avoid the challenges associated with modeling interacting particles. Moreover, it facilitates prediction and observation input processing directly in pixel space while providing strong inductive bias, including the sparsity of the dynamics resulting from the locality of the contact mechanics as well as the spatial equivariance of the dynamics. However, field-based representation is rarely explored in learning-based dynamics models and presents many design challenges, such as how to jointly represent states and actions, and compatibility with planning algorithms that rely on object-centric representations.

To this end, we introduce Neural Field Dynamics Model (NFD), a learning-based dynamics model for granular material manipulation. To account for the complex granular dynamics, we leverage the insight that the interaction between each granular object is dominated by local friction and contact, indicating that each voxel in the scene is only interacting with nearby voxels. To take advantage of such sparsity in the transition model and account for translation equivariance, we develop a transition model based on fully convolutional networks (FCN) that uses a unified density-field-based representation of both objects and actions. By using differentiable rendering to construct the density-field of actions, our model is fully differentiable, allowing it to be integrated with gradient-based trajectory optimization methods for planning in complex scenarios, such as pushing piles around obstacles.

We evaluate our approach with a variety of pile manipulation tasks in both simulation and the real world. We demonstrate that the field-based models are more accurate and efficient than existing latent dynamics [2] and particle-based methods [3]. Moreover, our model can solve unseen complex planning tasks, such as continuously pushing piles around obstacles. Additionally, we showcase the generalizability of our method by transferring a trained model to different environments with varying pusher and object shapes in a zero-shot setting.

## 2 Related Works

**Learned Dynamics Model with Inductive Bias.** Inductive biases, particularly object-centric representations, have been widely adopted in learning-based dynamics models. Particle-based representations serve as strong inductive biases for representing deformable objects [7, 3, 8, 4, 9, 10, 11, 12].In particular, DPI-Net [8] combines a hierarchical particle dynamics model with MPC-based control for deformable object manipulation. For planar pushing tasks, Xu et al. proposed an volumetric object representation learned via history aggregation [13]. However, particle-based approaches, also known as Lagrangian methods, face scalability issues as the number of particles increases, thereby making them computationally expensive and challenging to use in practical planning tasks. On the contrary, our proposed model employs a field-based representation, specifically the Eulerian method, which could overcome this limitation and also provide a strong inductive bias for learning.

**Piles Manipulation**, specifically moving piles of objects to target regions through pushing, is a challenging task due to complex dynamics and the large number of objects involved. Several special cases of this problem, including table wiping [14] (simpler dynamics) and multi-object pushing [15] (non-granular) have been proposed to be addressed by reinforcement learning. Suh et al. introduce a strong dense latent dynamics model baseline [2] for general piles manipulation problem. We show that our method achieves superior accuracy and sample efficiency by incorporating the field-based representation as model inductive bias. Imitation learning approaches such as Transporter Network [16] and Cliport [17] can directly predict the start and end poses of the pushes. However, these methods require task-specific oracle demonstrations and cannot easily generalize beyond training tasks. In contrast, our method trains on random interaction data and can solve unseen pile manipulation tasks in new environments. Recently, a concurrent work by Wang et al. [18] proposed a learning-based resolution regressor to address the scalability issues in particle-based dynamics models for pile manipulation. However, [18] require task-specific training data, for example, pile gathering and splitting require designated goal information during training, and hence making it challenging to generalize across unseen tasks. In contrast, our method does not rely on goal information during

training, and enables generalizations to new environments in zero-shot. Moreover, existing methods [2, 16, 14, 18] primarily focus on planning straight-line pushes and neglect the importance of obstacle avoidance in accomplishing tasks within complex real-world environments. In contrast, our proposed method employs trajectory optimization to plan curvilinear pushes, while considering geometric constraints such as obstacle avoidance.

**Optimization-based Trajectory Generation** Trajectory optimization plays a vital role in robotic applications as it enables robot agents to execute desired fluid movements while respecting environment constraints [19, 20, 21, 22, 23, 24, 25, 26, 27]. In particular, gradient-based methods [28, 19, 29, 30] can efficiently optimize trajectories to minimize cost while satisfying differentiable dynamics and constraints. We incorporate a differentiable action rendering module into our unified field-based dynamics model, thereby making it entirely differentiable end-to-end and compatible with a gradient-based trajectory optimizer.

# 3 Method

Developing dynamics models for granular materials manipulation is challenging due to the complex non-convex and non-linear nature of their dynamics. In addition, developing models that are applicable to real-world scenarios presents practical challenges; specifically, ensuring that the model's complexity is scalable and independent of the number of particles involved is crucial. Furthermore, the dynamics model must be amenable to planning algorithms, such as trajectory optimization, in order to perform complex tasks in real-world scenarios where obstacles exist.

To address these challenges, we proposed Neural Field Dynamics (NFD) based on Fully Convolutional Network (FCN). The FCN model, which brings strong inductive bias, is well-suited for capturing the complex nature of granular materials dynamics. Moreover, by using the Eulerian approach that represents objects as images, our NFD models show better scalability compared to the Lagrangian approach that models objects as particles, further enhancing their potential for use in real-world scenarios. In this section, after describing the formulation of the problem, we explain why our proposed model is able to better capture the complex nature of granular materials dynamics, and we also discuss how it can be integrated into trajectory optimization algorithms.

**Problem statement.** The goal is for a robot to use a flat-surface pusher, such as a spatula, to interact with piles of granular materials in order to push all the particles to a designated target region. Our model operates on a quasistatic system where a state is represented as a density field image of the pile $s \in \mathcal{R}^{H \times W}$, and the target region, denoted as $G \in \mathcal{R}^{H \times W}$, is a 2D binary mask representing the goal region where all the piles are intended to be gathered. Unlike existing methods that only consider the start and end positions of a straight-line push, our method generates curvilinear trajectories $\tau = (x_1, x_2, ...x_H)$, where $x_i \in SE(2)$ is the top-down 2D pose of the pusher, to support more flexible behaviors, allowing the robot to execute continuous pushes satisfying specific geometric constraints such as avoiding obstacles.

## 3.1 State and Action Representation

**Density field state representation.** To develop a scalable model that is independent of the number of particles, we propose representing the system state as a grid-based density field of the granular materials. This approach, known as the Eulerian approach, avoids the explicit modeling and perception of each particle and enables our model to scale effectively in complex real-world scenarios. Specifically, we capture the density field state $s$ by segmenting an RGB image into a one-channel occupancy grid after an orthographic projection.

**Differentiable rendering for field-based action representation.** To allow our FCN-based dynamics model (Sec. 3.2) to directly capture the local interaction between the pusher and the objects, we propose mapping the pushing action into a similar spatial field-based representation. We implement the mapping function using a differentiable rasterizer, allowing gradients from the dynamics model to backpropagate to the trajectory, enabling gradient-based optimization (see Sec. 3.2). Con-

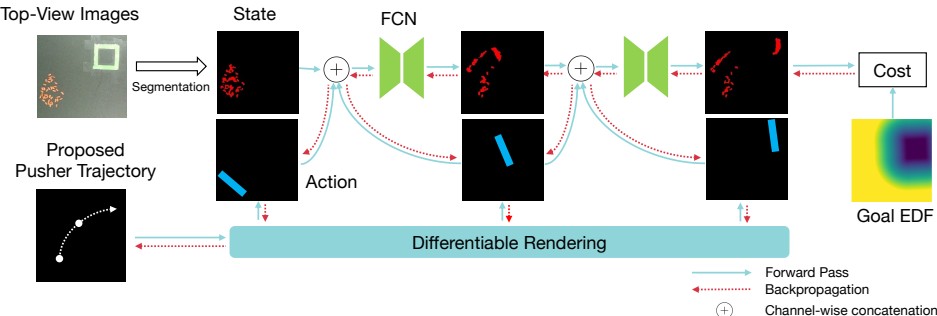

Forward Pass
Backpropagation
+ Channel-wise concatenation

Figure 2: We propose Neural Field Dynamics (NFD) model, a fully convolutional network that employs a unified density-field-based representation of both object states and actions. By using differentiable rendering, the dynamics model is fully differentiable, and enables us to integrate learned field dynamics with gradient-based trajectory optimization methods.

cretely, we assume linear pusher motion during a short time interval $[t, t+1)$ and represent an action as $\boldsymbol{a}_t := [r(\boldsymbol{x}_t), r(\boldsymbol{x}_{t+1})]$, where $\boldsymbol{x}_t \in SE(2)$ is the proposed pusher pose at time $t$ and $r : SE(2) \to \mathcal{R}^{H \times W}$ is a differentiable rendering function (see appendix for more details) that rasterizes a pusher pose into a one-channel image representing the density field of the pusher in the plane. A sample rendered trajectory is illustrated in Fig. 2

## 3.2 Learning Localized Field Dynamics

We propose to use Fully Convolutional Networks (FCN) as the backbone for the dynamics model to (1) improve sample efficiency of the dynamics model and (2) incorporate strong inductive bias of the dynamics by exploiting the localized dynamics of granular material manipulation. Given the current state $\boldsymbol{s}_t$ and the proposed action $\boldsymbol{a}_t$, both in field representations, the model captures the forward dynamics $\hat{\boldsymbol{s}}_{t+1} = f_\theta(\boldsymbol{s}_t, \boldsymbol{a}_t)$, where $\boldsymbol{s} \in \mathcal{R}^{H \times W}$ is the state of the objects, $\boldsymbol{a}_t := [r(\boldsymbol{x}_t), r(\boldsymbol{x}_{t+1})]$, $\boldsymbol{x}_t \in SE(2)$ is the proposed pusher pose at time $t$ and $r : SE(2) \to \mathcal{R}^{H \times W}$ is the differentiable rendering function.

FCN-based dynamics models are equivariant to translation by nature [31], as $f_\theta(\omega * \boldsymbol{s}, \omega * \boldsymbol{a}) = \omega * f_\theta(\boldsymbol{s}, \boldsymbol{a})$, where $f_\theta$ is the FCN with parameter $\theta$, and $\omega$ is a translation. Such equivariance to translation has shown to provide better learning sample efficiency [16].

We further conducted an in-depth analysis to justify why FCN incorporates strong inductive bias. Ideally, a neural network model with a strong inductive bias should capture the intrinsic physical properties of a system even in the absence of training. For tasks involving granular material manipulation, a fundamental physical property is the localized dynamics. This means that the end-effector or particles only interact with neighboring particles. By drawing an analogy from the correlation function in statistical physics [32], such localized dynamics can be described by evaluating the correlation between variations in the pusher density field ($\dot{\boldsymbol{a}} := \frac{dr(\boldsymbol{x})}{dt}$) at one point and the variations in the object density field ($\dot{\boldsymbol{s}} := \frac{d\boldsymbol{s}}{dt}$) at another. This correlation changes based on the distance between these points. In scenarios where objects interact locally, higher correlations are anticipated when two points are in proximity, and vice versa. By evaluating this metric, which characterizes

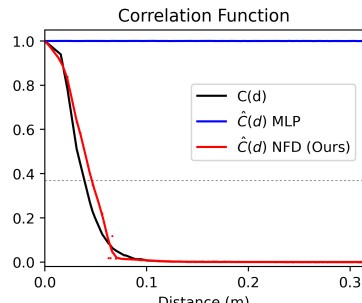

Figure 3: Correlation functions of the piles pushing dataset $C(d)$, as well as FCN's $\hat{C}(d)$ and MLP's $\hat{C}(d)$. The closeness to the black line indicates the model's capacity to adapt to the data. The dashed line corresponds to $1/e$ and determines the correlation length (see Appendix for more details).

the locality of dynamics, in both the dataset and the FCN model, we found that the FCN model displayed significant similarities with the dataset even without training, as is shown in Fig. 3 This

suggests that the FCN model inherently possesses a strong inductive bias towards granular material dynamics represented as density fields. We provide a detailed explanation of the correlation functions in the Appendix B.1.

Our dynamics model adopts a shallow U-Net [33] architecture to ensure computational efficiency. The network is trained in a self-supervised manner, using randomized pushing data in environments to predict subsequent steps following an action, minimizing the loss function $\mathcal{L}_{train} = ||f_\theta(s_t, a_t) - s_{t+1}||_F^2$ between the predicted and observed states, where $|| \cdot ||_F$ is the Frobenius norm. To perform sequential prediction for long-horizon, the predicted state $\hat{s}_{t+1}$ is fed again into the dynamics model along with the proposed action $a_{t+1}$. Note that $a_t, a_{t+1}$ are not necessarily along the same direction, and hence allow us to model curvilinear trajectories.

### 3.3 Trajectory Optimization with Learned Dynamics

Because our learned dynamics is end-to-end differentiable, it can be directly integrated with gradient-based trajectory optimization to effectively handle complex scenarios involving geometric constraints, such as collision avoidance. The objective of such optimization is to maximize the amount of material that is pushed to the designated target region, while simultaneously avoiding collisions with obstacles. The optimization problem is defined as follows:

$$\min_{\boldsymbol{x}_t \in SE(2)} \ell_{goal}(\boldsymbol{s}_T; \boldsymbol{G}) + \sum_{i=0}^{T} \ell_{action}(\boldsymbol{x}_i; \boldsymbol{O}) + \sum_{i=1}^{T} \ell_{state}(s_i; \boldsymbol{O}) \tag{1}$$
$$\text{s.t. } \boldsymbol{s}_{t+1} = f_\theta(\boldsymbol{x}_t, [r(\boldsymbol{x}_t), r(\boldsymbol{x}_{t+1})]) \; \forall t \in [0, T-1]$$

where $\boldsymbol{G} \in \mathcal{R}^{H \times W}$ is the mask of the target zone and it is assigned 0 inside the zone and 1 out of the zone. The individual objective functions are defined as follows:

$$\ell_{goal}(\boldsymbol{s}_T; \boldsymbol{G}) := \alpha_1 \langle EDF(\boldsymbol{G}), \boldsymbol{s}_T \rangle_F - \alpha_2 \langle \boldsymbol{G}, \boldsymbol{s}_T \rangle_F$$
$$\ell_{state}(\boldsymbol{s}_i; \boldsymbol{O}) := \alpha_3 \langle \boldsymbol{O}, \boldsymbol{s}_T \rangle_F \tag{2}$$
$$\ell_{action}(\boldsymbol{x}_i; \boldsymbol{O}) := \alpha_4 \langle \boldsymbol{O}, r(\boldsymbol{x}_i) \rangle_F$$

where $EDF(\boldsymbol{G})$ computes the Euclidean Distance Field (EDF) of the target zone. $\langle, \rangle_F$ is the Frobenius inner product (i.e. the sum of element-wise product), $\alpha_1, \alpha_2, \alpha_3, \alpha_4$ are positive weight parameters. In our experiments, we let $\alpha_1 = 1, \alpha_2 = 2, \alpha_3 = 1, \alpha_4 = 2$. Moreover, $\ell_{action}$ and $\ell_{state}$ are loss functions utilized to address geometric constraints such as obstacles avoidance, and $\boldsymbol{O} \in \mathcal{R}^{H \times W}$ is the mask of the obstacle objects. The objective is flexible and can handle a variety of tasks, including including piles splitting (where piles are pushed towards multiple target regions while ensuring balanced distribution) and piles spreading, without the need for explicit training for these specific scenarios. For further information regarding the optimization objectives for piles splitting and spreading, please refer to the Appendix.

**Generating curvilinear trajectories.** Most existing pile manipulation methods are limited to generating straight pushes. However, curved trajectories are required for complex scenarios such where straight push is not possible (e.g., blocked by obstacles) or suboptimal. We enable curvilinear trajectory generation by parameterizing the trajectory with B-spline. The parametrized curves could be viewed as an additional constraint in Eq.1. We include additional detail in Appendix. To further account for the non-convex nature of the dynamics model and mitigate the risk of local minima, we perform an initial random sampling of the control points for the spline curves. Subsequently, the optimization is conducted in batches, and the resulting optimal solution that minimizes the cost function is chosen as the output of the planning model.

## 4 Experiments

We conduct experiments to validate that our field-based dynamics model is more accurate and efficient than existing latent and particle-based methods in capturing granular material dynamics. Moreover, we show that our model can solve unseen pile manipulation tasks such as pile splitting and

spreading, as well as pile pushing around obstacles. Finally, we transfer a trained model to new scenarios, including unseen object and pusher shapes and a physical robot setting.

**Simulation setup.** The Pybullet simulation environment is adapted from Ravens [16]. Throughout the data generation process and all evaluations, unless explicitly stated otherwise, we use a set of 50 cubic blocks with a size of 1cm and a planer pusher of length 5cm.

**Physical robot setup.** The robot setup consists of a Franka Panda manipulator and a Intel RealSense camera (see Appendix for more details). The camera captures a top-down view of environment. The captures images are rectified using homographic warping, followed by color and depth thresholding to extract the object density field. We directly transfer a model trained in simulation for real-world experiments. To ensure consistency between the simulation and real-world settings, we resize the input image to capture a workspace same size as in simulation. The pusher is of identical width as the simulated pusher and is affixed to the robot gripper.

**Evaluation setup.** For offline analysis of dynamics model, we measure prediction performance using the Mean Squared Error (MSE) between the predicted and the ground truth density field image. We measure rollout performance with success rate, which is the fraction of objects that are inside the goal region at the end of an episode, and cost function $\ell_{eval}(\boldsymbol{s_t}; \boldsymbol{G}) = \langle EDF(\boldsymbol{G}), \boldsymbol{s_t} \rangle_F$ which could also be viewed as the Control Lyapunov Function as has been pointed out by Suh et al [2]. The episode length is 10 steps (pushes). The success rates are averaged across 20 trials with random initial state and goal regions. This evaluation reflects the objective of efficiently pushing piles into the target region with a minimal number of pushes.

**Baselines.** The baseline methods for pile manipulation can be categorized into three groups: (1) field-based latent dynamics model, (2) particle-based dynamics model, (3) and model-free imitation learning. We pick the strongest among each as our baseline, with a brief description for each below.

- **DVF [2]** is a strong dynamics model that uses field-based state representation without inductive biases. The original DVF only predicts the start and end position of a straight push (*Single Pred.*). To facilitate fair comparison, we evaluated DVF for both single and sequential prediction (*Seq. Pred.*). We also augment DVF with the same gradient-based optimizer used by NFD, despite the original work only showed shooting-based planning.
- **DPI-Net [3]** is a particle-based method built on GNNs. We adapted the original codebase [3, 34] for piles manipulation tasks. Unlike field-based methods, the method has access to the ground truth particle position for rollout experiments in simulation. Moreover, since the output of DPI-Net is particle positions, the image-based observation is obtained by rendering the predicted position of each particle during prediction evaluation.
- **Object-centric (OC) [2]** is a non-learning method proposed in [2] that approximates the particle pushing dynamics with analytical model.
- **Transporter Network [16]** is a model-free method trained with expert demonstrations. It adopts a similar FCN backbone as our method, but instead of predicting dynamics, it directly outputs the next pushing action.

We use 256 planning samples for all model-based methods. For our NFD mdoel, we evaluate 3 different planning strategies: (1) straight line pushing using the random shooting (RS) method [35] with only forward pass (NFD-RS), (2) trajectory optimization with straight line pushing (NFD-Opt), and (3) trajectory optimization with curved pushing (NFD-Curve).

### 4.1 Main experiment results

**Our method outperforms the baseline planning methods in accuracy and speed.** Quantitative results (Tab. 1) and qualitative results (Fig. 4) demonstrate that our method outperforms the latent-dynamics DVF model in prediction accuracy for both single-push and sequential-push scenarios, showing the advantage of the FCN inductive bias. Additionally, our method is far more computationally-efficient than the baseline methods, as shown in Tab. 2, which quantifies the GFLOP required for each prediction. For rollout performance, our sequential prediction-based ap-

proach outperforms all baselines in both relative cost reduction and the task success rate (see Tab. 1 and 3). Notably, despite our method not being trained on expert data, it achieves better performance than the Transporter Network which was trained by 10,000 expert pushes. As illustrated in Fig. 5, our method not only reaches the goal significantly faster than other methods but also closely approximates expert behavior, even in the absence of expert training data. However, in the second half of the episode, our method's convergence rate is slower than that of expert demonstration. This is because our randomly generated dataset rarely involves pushing one particle to a pile of particles once most particles are already in the target region, which results in less accurate predictions.

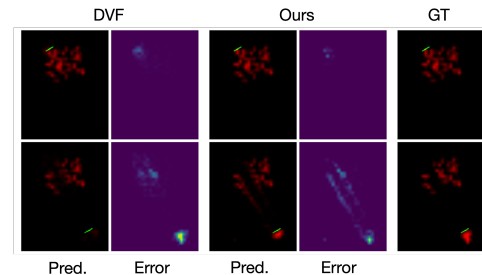

(a) MSE Prediction Error Comparison

|              | Single Pred. | Seq. Pred. |
|--------------|--------------|------------|
| DVF [2]      | 7.80E-04     | 2.13E-03   |
| NFD (Ours)   | **5.72E-04** | **4.42E-04** |

(b) Visualizing model predictions and errors.

Figure 4: Quantitative (a) and qualitative (b) comparison between field-based methods.

Table 1: Rollout Performance Comparison of Model-based Methods

|                              | Model        | Type  | Cost    | Success |
|------------------------------|--------------|-------|---------|---------|
| Particle-based               | DPI-Net [3]  | RS    | 0.448   | 0.125   |
|                              | DPI-Net [3]  | Opt   | 0.2325  | 0.449   |
|                              | OC [2]       | RS    | 0.1265  | 0.459   |
| Image-based (Single Pred.)   | DVF [2]      | RS    | 0.1838  | 0.606   |
|                              | DVF [2]      | Opt   | 0.1498  | 0.768   |
|                              | NFD (Ours)   | RS    | 0.1677  | 0.744   |
|                              | NFD (Ours)   | Opt   | 0.07173 | 0.899   |
| Image-based (Seq. Pred.)     | DVF [2]      | RS    | 1.103   | 0.014   |
|                              | DVF [2]      | Opt   | 1.126   | 0.015   |
|                              | NFD (Ours)   | RS    | 0.0819  | 0.8     |
|                              | NFD (Ours)   | Opt   | **0.0197** | **0.966** |
|                              | NFD (Ours)   | Curve | 0.0416  | 0.936   |

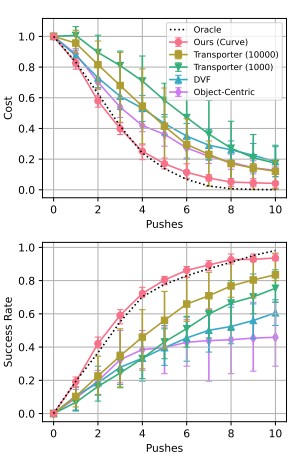

Figure 5: Detailed comparison of rollout performance.

Table 2: Computation Cost Comparison among Model-based Method

|            | Params    | GFLOP 50 particles | GFLOP 200 particles |
|------------|-----------|--------------------|---------------------|
| DPI [3]    | 3.91E+05  | 7.53E-01           | 8.43E+00            |
| DVF [2]    | 6.95E+07  | 1.62E-01           | 1.62E-01            |
| NFD (Ours) | **1.53E+04** | **7.36E-03**    | **7.36E-03**        |

**Our method can generate flexible pushing trajectories in challenging environments**. Notably, our approach can effectively plan in the presence of obstacles between the initial objects and the target region. As illustrated in Tab. 3, our method achieves high performance despite not being explicitly trained for such scenarios. Furthermore, we demonstrate that our model can be applied to different tasks including splitting and spreading without additional training (see Appendix for details). This highlights the generalizability of our approach, especially compared to model-free methods like the Transporter, qualitative results are shown in Fig. 6.

Table 3: Comparison with Model-free Method

|  | # Expert Pushes | w/o Obstacle | | w/ Obstacle | |
|---|---|---|---|---|---|
|  |  | Cost | Success | Cost | Success |
| Transporter [16] | 1000 | 0.1731 | 0.754 | 0.575 | 0.049 |
|  | 10000 | 0.1211 | 0.835 | 0.5331 | 0.088 |
| NFD-Linear (Ours) | 0 | **0.0197** | **0.966** | 0.3820 | 0.455 |
| NFD-Curve (Ours) | 0 | 0.0416 | 0.936 | **0.1318** | **0.838** |

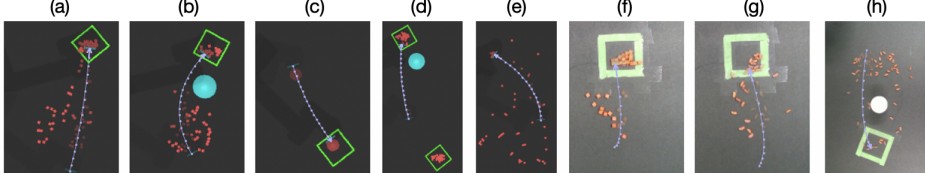

Figure 6: **Simulated**: (a) curved pushing (b) obstacle avoidance (c) pushing larger object (d) pile splitting (e) pile spreading. **Real-world**: (f) pushing blocks and (e) beans (h) obstacle avoidance. **Our method can generalize to diverse environments with different dynamics**. As we varied the shapes and sizes of objects and pushers, we notice that despite using unseen pusher lengths (50% longer and 20% shorter) and unseen objects (one big circular disk and 200 small octahedrons instead of small cubic blocks), our method still achieved comparable performance (see Tab. 4). Moreover, in real-world experiments, our model performs well with unseen objects, such as beans, without retraining (see Fig. 6 and Tab. 4).

## 5 Limitation

**Suboptimal solutions.** The non-convex nature of the dynamics may lead to suboptimal trajectory solutions. This often happens when pusher fails to interact with any particles and the gradient becomes zero. This issue is particularly pronounced when dealing with scenarios with isolated particles in specific regions, necessitating a larger number of samples to find the optimal trajectory. Such limitation could be addressed by combining our model with a model-free method, such as Transporter, and using the output of the model-free approach as initialization of the trajectory optimizer.

Table 4: Generalization Results - Rollout Evaluation under Diverse Environment without Training

|  | Case | Rollout Cost |
|---|---|---|
| Piles (Training) | 50 blocks | 0.0416 |
| Different object shape | 200 small-octahedrons | 0.0905 |
|  | 1 large disk | 0.0003 |
| Different Pusher shape | Larger pusher | 0.0156 |
|  | Smaller pusher | 0.1179 |

**Myopic planning.** While effective, our method greedily optimizes for the next best trajectory given a global cost function. However, in complex scenarios, it may be necessary plan ahead multiple pushes to determine the next best action. Our future work will investigate longer-horizon prediction and develop a more efficient sampling and optimization strategy to enable long-horizon planning.

## 6 Conclusion

We presented Neural Field Dynamics Model (NFD), a learning-based dynamics model that effectively addresses granular material manipulation challenges. Leveraging field-based representations and fully convolutional networks (FCN), our model outperforms existing methods in both accuracy and efficiency. Its ability to perform complex tasks and adapt to varying environments highlights its robustness and versatility, promising significant improvements in granular object manipulation.

**Acknowledgments**

This work has taken place in the Robot Learning and Reasoning Lab (RL2) at Georgia Institute of Technology. RL2 research is partially supported by NSF (2101250). We thank Frank Dellaert, Fan Jiang, Yetong Zhang and Gerry Chen for insightful discussions, and the anonymous reviewers for their comments and feedback on our manuscript.

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

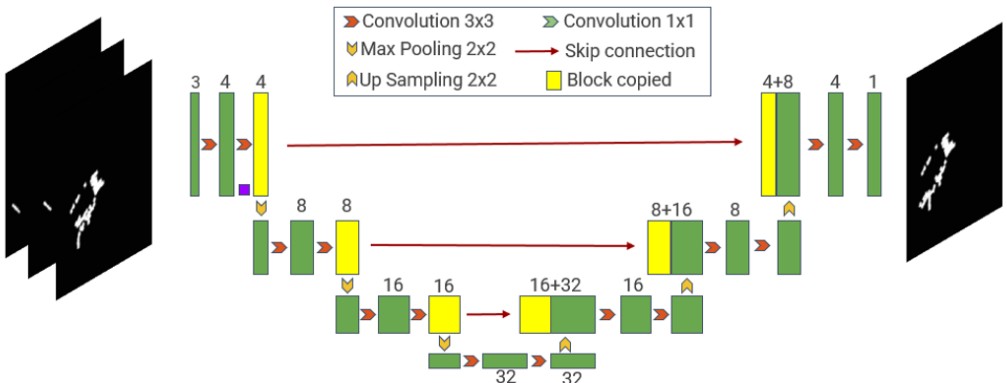

Figure 7: **Architecture of proposed neural network**

# A    Model Details

Our learned dynamics model takes current state in image and action in poses as input. To bridge field-based state representation and object based pose representation, we employed differentiable rendering to transform pusher poses from $SE(3)$ to $\mathcal{R}^{H \times W}$. In this context, we utilized a simplified SoftRasterizer[36], since our scenario involves solely 2D motion without occlusion and texture. The rendered image could be differentially obtained by transforming the probability map of the pusher in the 2D plane.

To ensure the smoothness of the generated trajectory, the trajectory of the pusher is constrained to be a B-Spline curve, which is defined as $C(u) = \sum_{i=0}^{m} N_{i,n}(u) \cdot P_i$, where $P_i$ is the control points, $C(u)$ is the pusher position on the curve at parameter $u$ , $N_{i,n}$ is the $i$-th basis function at degree $n$. Noticed that the basis function evaluated at $u$ could be pre-computed, enabling efficient computation of $C(u)$ using matrix multiplication given $P_i$. This allows us to directly optimize for the control points $P_i$ for trajectory planning. In our experiments, we used Bazier curve, a special case of the B-Spline curve, while our method can readily extend to handle more complex scenarios with B-spline curves.

## A.1    Objective Function for different tasks

Our learned field dynamics model exhibits versatility beyond the task of gathering particles to a target position. It can be applied to various tasks, including splitting piles and spreading piles. When splitting piles, the objective is to push the piles towards multiple designated target regions while ensuring an equal distribution of particles among these targets. To achieve this, we utilized a goal mask $\boldsymbol{G}$ where 0 is assigned to objects inside any one of the N targets. By computing the minimum value of N single-target EDF, we obtained the EDF for the N-target and could similarly applied the objective function described in Equation 2. To evenly split the target into different goals, we compute the Voronoi diagram of $G$ which is a partition of a plane into regions close to each of the given target regions, then the additional objective term objective function $\ell_{split}(\boldsymbol{s}_i; \boldsymbol{G})$ simply computes the standard deviation of the weight of particles distributed within each partition of the Voronoi diagram.

In the case of piles spreading, we partitioned the space into low-resolution grids. The objective function $\ell_{spread}(\boldsymbol{s}_i)$ was introduced to quantify the standard deviation of the total weight of particles distributed within each grid. This measure enabled us to assess the uniformity of particle distribution across the workspace.

# B NFD with Fully Convolutional Network Backbone

The architecture of our proposed neural network model is shown in Fig.7. It is a shallow version of U-Net[33], this is inspired by the fact that the dynamics of granular particle interation is localized, and hence we do not need a very deep network to capture long range interaction. We performed correlation analysis on our dataset and the model to justify this.

## B.1 Details on Spatial Correlation Analysis

In this section, we investigated the correlation function between variations in the pusher density field ($\dot{\boldsymbol{a}} := \frac{dr(\boldsymbol{x})}{dt}$) at one point and the variations in the object density field ($\dot{\boldsymbol{s}} := \frac{d\boldsymbol{s}}{dt}$) at another. It measures how two positions in the field affect each other as a function of distance. We perform evaluations of the correlation function both in the dataset and the learned dynamics models, to justify that the FCN model inherently possesses a strong inductive bias towards granular material dynamics, compared with a fully connected network.

Firstly, we define the correlation between the variation in pusher density field $\dot{\boldsymbol{a}}$ of pixel $i$ and the variation in particle density field $\dot{\boldsymbol{s}}$ of pixel $j$:

$$\rho_{ij} = \frac{Cov(\dot{a}_i, \dot{s}_j)}{\sigma_{\dot{a}_i} \sigma_{\dot{s}_j}}$$

noticed that $\sigma_{\dot{a}_i}, \sigma_{\dot{s}_j}$ is the variance of $a_i, s_j$ respectively and could be dropped as we assume they are normalized and translational equivariant. We could then compute the correlation function from the dataset:

$$C(d) = E\left[\rho_{ij} | D(i,j) = d\right] \tag{3}$$

where $D(i,j)$ is the Euclidean distance between pixel $i$ and $j$. $C(d)$ represents the correlation between the variation of the pusher density field and the variation of the particle density field as a function of distance, characterizing the localized dynamics in the dataset.

Secondly, we compute the correlation function of a dynamics model $f_\theta$. As a result of regression toward the mean [37], and assuming $\dot{a}_i$ and $\dot{s}_j$ are correlated linearly, we have $\Delta \dot{s}_j = \rho_{ij} \Delta \dot{a}_i$, indicating if $\dot{a}_i$ changes by $\Delta \dot{a}_i$, the expected change in $\dot{s}_j$ is $\rho_{ij} \Delta \dot{a}_i$, and hence $\rho_{ij} = E\left[\frac{\partial \dot{s}_i}{\partial \dot{a}_j}\right]$. Therefore we could similarly obtain the correlation $\hat{\rho}_{ij}$ of a dynamics model by:

$$\hat{\rho}_{ij} = E\left[\frac{\partial \hat{\dot{s}}_i}{\partial \dot{a}_j}\right]$$

which represents how much $\hat{\dot{s}}_i$ changes by expectation, for a small change in $\dot{a}_i$, where $\hat{\dot{s}}^{(t)} = \hat{\boldsymbol{s}}_{t+1} - \boldsymbol{s}_t$. We could accordingly compute $\hat{\rho}_{ij}$ through back-propagation of the neural network. Thus the correlation function of the dynamics model could be written as:

$$\hat{C}(d) = E\left[\hat{\rho}_{ij} | D(i,j) = d\right] \tag{4}$$

where $\hat{C}(d)$ characterized the localization of the dynamics model, specifically the extent to which altering an action will impact the output state located a distance $d$ away from it.

To calculate the correlation function $C(d)$, we consider all pairs of pixels (state-action) in neighboring timesteps by evaluating $\dot{s}_i \dot{a}_j$. These pairs are then grouped based on their Euclidean distance, and the average is computed to obtain the correlation function $C(d)$. For $\hat{C}(d)$, we perform back-propagation on the neural network to compute $\frac{\partial f_\theta}{\partial r(\boldsymbol{x}_t)}$ for all state-action pixel pairs. This computation is conducted using random inputs and random network weight initialization. Similar to $C(d)$, the results are grouped based on the Euclidean distance between each pair, and the average is taken to obtain $\hat{C}(d)$.

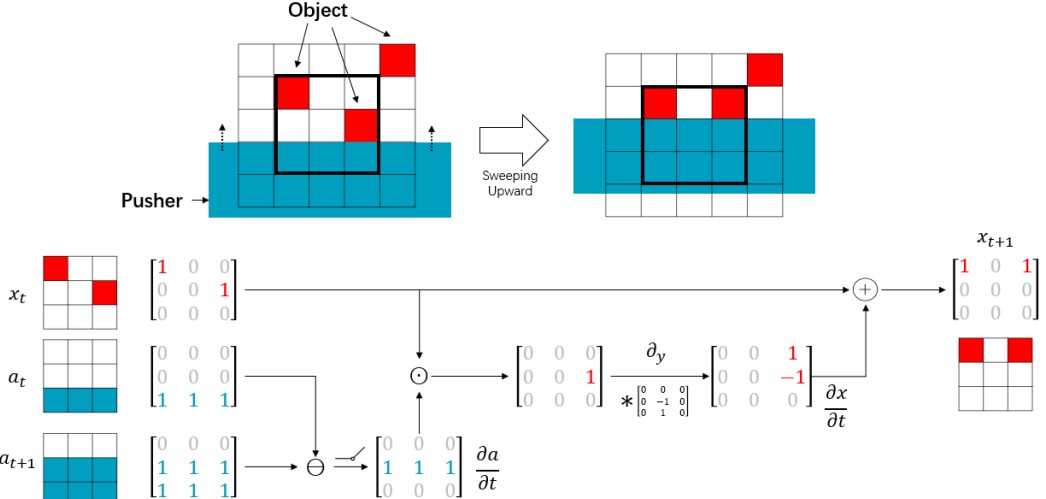

Figure 8: **Toy example of a simplified scenarios**

The computation results of $C(d)$ for our dataset using Eq. 3 are shown as the black curve in Fig. 3, indicating that the correlation strengthens as the distance between pixels decreases. This suggests that when two pixels are in close proximity, a change in one pixel is more likely to influence the other, which can be attributed to localized dynamics. Additionally, we evaluate $\hat{C}(d)$ for an untrained FCN model. The red curve in Fig. 3 shows that even without training, the behavior of $\hat{C}(d)$ in the FCN is similar to that of the dataset. Specifically, the correlation diminishes as the distance grows, implying that the FCN can effectively capture the localized nature of the dynamics. This suggests that the FCN inherently possesses a strong inductive bias towards the dynamics system. On the other hand, the correlation for an MLP model, represented by the blue curve in Fig. 3, remains consistent regardless of distance. This indicates that a change in one pixel is equally likely to influence all other pixels, irrespective of their distance apart. Such behavior does not align with the localized nature of the dynamics, suggesting that the MLP lacks the necessary inductive bias for manipulating granular materials. Additionally, we observed a good fit between our correlation function $C(d)$ and $exp(-\frac{d}{\xi})$, which is a commonly used function for correlation in statistical physics[32]. This fitting allows us to estimate the correlation length $\xi$ of the system, a parameter often used to characterize system localization. Notably, the correlation length of our dataset is approximately 0.035, which closely aligns with the correlation length of the FCN. In contrast, the correlation length of the MLP tends towards infinity. This indicates that the FCN model is more suitable for modeling granular material manipulation with localized dynamics.

## B.2  Toy Example

In addition, we present a toy example illustrating the effectiveness of a shallow FCN with field-based action representation in modeling the dynamics of granular materials in a simplified setting. As depicted in Fig. 8, the blue area represents the pusher, while the red area represents the particles. We made the assumption that the pusher only moves upward by 1 pixel per step and neglected particle-particle interactions. Remarkably, even with this minimalistic approach, we could see that the shallow FCN successfully captured the dynamics of this scenario, showcasing its capability in modeling such cases.

## C  Supplementary Experimental Results

### C.1  Experiment Details

In the roll-out experiments and data generation process, we use 50 blocks with dimension 1cm×1cm. In roll-out experiments, the size of the target region is 12cm × 12cm and is randomly sampled away from the particle's initial positions.

### C.2  Additional Qualitative Results

We present further qualitative results in Fig. 9 for two representative cases: one using a linear planner and the other using a curved planner.

### C.3  Additional Results on Trajectory Optimization

Fig. 10 displays representative examples of trajectory optimization. The figure illustrates that, despite the initial trajectory being suboptimal, the optimization process refines it, leading to trajectories with reduced costs.

### C.4  Ablation Study on Action Representation

To demonstrate the effectiveness of the field-based action representation, we replace DVF's vector-based action with the field-based action representation (Sec. 3.1). Tab. 5 shows the substantial improvements in prediction and rollout performance achieved through the utilization of field-based action representation, particularly in scenarios with limited training data. Note that our proposed field-based action representation is not limited to pile manipulation and has the potential to be used in many model-based learning methods in robotics.

Table 5: Ablation - Action Representation and FCN

|  | Error (Sequential Prediction) | | | Loss (Rollout) | | |
|---|---|---|---|---|---|---|
|  | 2000 | 6000 | 20000 | 2000 | 6000 | 20000 |
| DVF [2] | 3.98E-03 | 1.81E-03 | 2.13E-03 | 1.088 | 1.06 | 1.103 |
| DVF - Improved | 1.43E-03 | 6.77E-04 | 5.67E-04 | 0.4155 | 0.1727 | 0.1001 |
| Ours | **6.97E-04** | **4.79E-04** | **4.42E-04** | **0.1348** | **0.0993** | **0.0819** |

## D  Real Robot Setup

We include a picture (Fig. 11) of the real robot setup as well as the pusher, the objects (beans), and the obstacles (can) used in the experiments.

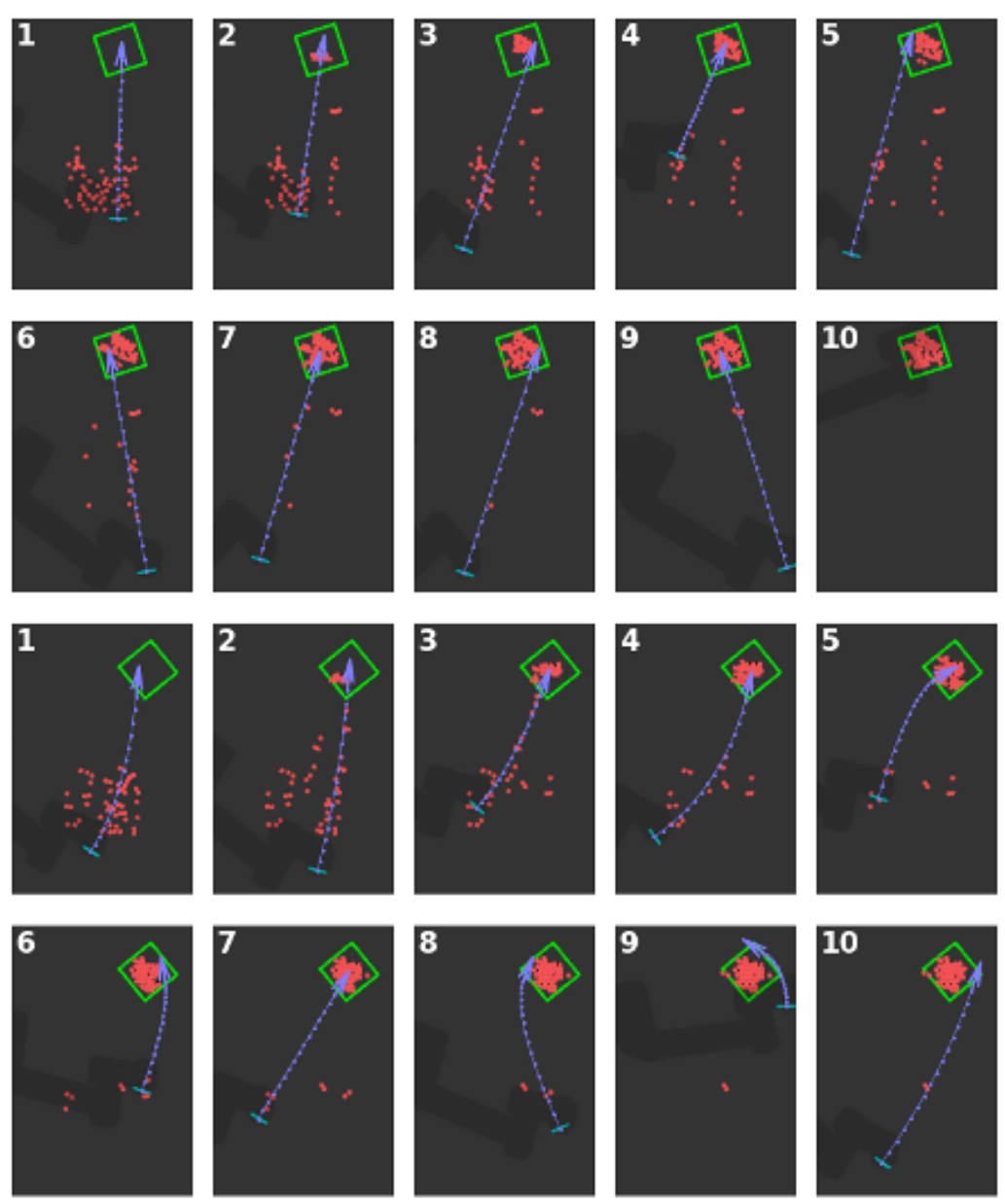

Figure 9: Qualitative rollout examples in simulation using (up) a linear planner and (bottom) a curved planner.

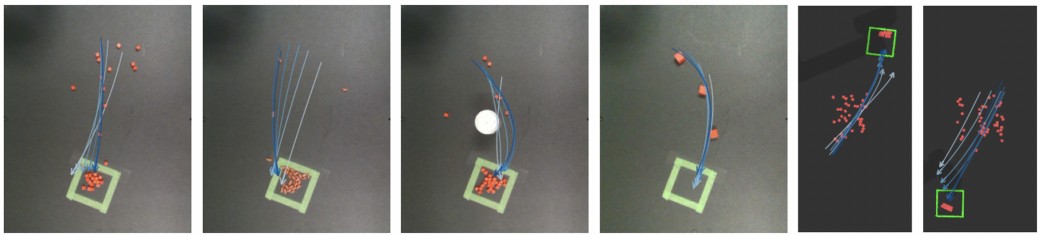

Figure 10: Visualization of the trajectory optimization process. White curves represent the initial trajectory, and the darkest curves highlight the optimized trajectory. The gradations in color (white to blue) indicate the iteration steps throughout the optimization.

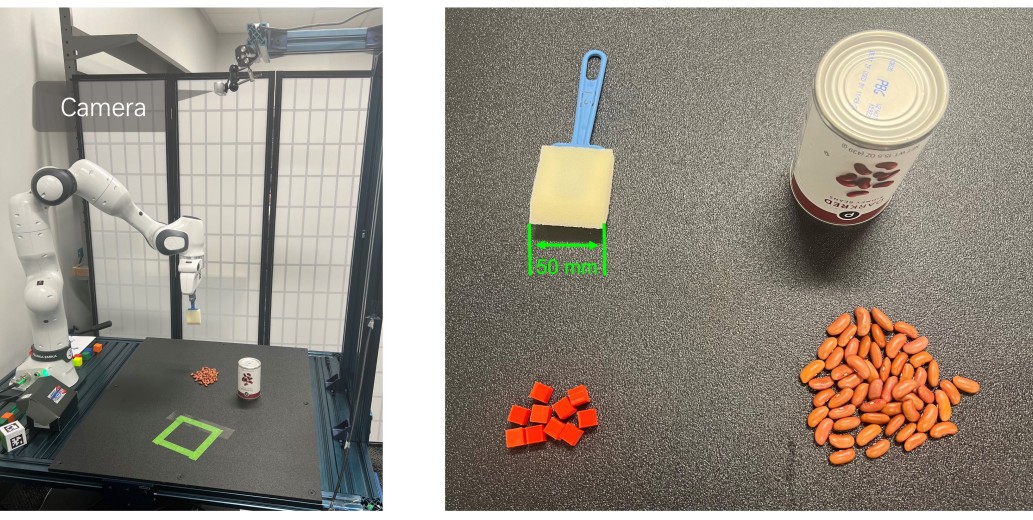

Figure 11: (Left) Real robot setup. (Right) Pusher, objects, and obstacles used in the experiments.

