# OpenReview forum: "Neural Field Dynamics Model for Granular Object Piles Manipulation"
_robot-learning.org/CoRL/2023/Conference — CoRL 2023 Poster_

### Official Review · Reviewer_9WbQ · 2023-06-23

**Confidence:** 3
**Originality:** Good
**Technical Quality:** Good
**Clarity Of Presentation:** Good
**Impact:** 3

**Recommendation:**

Weak Accept: I recommend accepting the paper, but will not argue for my recommendation if the majority of other reviewers have a different opinion.

**Review:**

The obstacle avoidance with moving granular objects is a novel problem introduced in this paper.

The paper is technically sound, with clear explanations for trajectory optimization with learned dynamics. When introducing the model, reference the appendix for model details. Otherwise, at a first glance of the paper, it seems like there are no model details.

Trajectory optimization is very clear and well explained but the correlation equation is hard to understand as there is little explanation. Results and experiments are significant and well explained.

The limitations of the work are well explained and I did not find many other drawbacks on the work.


**Quality Of The Limitations Section:**

Limitations are addressed clearly

**Questions For Rebuttal:**

1. In the abstract, it states the proposed method “significantly exceeds” existing methods. List percentage improvement.

2. In the discussion on Figure 3, it is unclear why the MLP-based dynamics model has high correlation even as the pusher-object distance increases. The paper states it is because it has poor matching with the dataset correlation. If that is the case, shouldn’t there be low correlation, not high? Also, it is unclear what pusher-object distance is.

3. In the baselines section, which of the 3 groups of baselines does object-centric (OC) belong to? It would be much easier for the reader if each baseline has (1), (2), or (3) listed in it.

4. Can you use linear NFD to do obstacle avoidance, through piecewise linear motions?

5. There are a few typos. 3.2: that as poor → that has poor. 3.3: including including → including. Exp, physical robot setup: captures → captured. Exp, Baselines: mdoel → model. Limitations, myopic planning: necessary plan → necessary to plan.


**Robotics Focus:**

Sufficient demonstration on hardware

**Summary Of Paper:**

This paper introduces Neural Field Dynamics Model which uses a network to obtain a density representation of piles of granular objects. The action rendering module also proposed in this work is fully differentiable and can be used for learning curvilinear trajectories, rather than only linear trajectories, which is a limitation of prior work. Curvilinear trajectories advance this work and allow it to perform obstacle avoidance when moving granular objects to target locations.

**Summary Of Recommendation:**

I give this paper a weak accept over a strong accept due to lack of clarity in the discussion of how correlation is calculated, little explanation for the result shown in Figure 3, and the fact that the baselines were hard to understand. It is an accept over a reject as trajectory optimization is well explained and novel in that it allows curve path planning for object avoidance when manipulating granular objects. Results also show NFD-curve does outperform the model-free baseline. Overall, in model based method comparison, NFD linear does outperform baselines in Table 1.

---

### Official Review · Reviewer_Un8n · 2023-07-18

**Confidence:** 3
**Originality:** Very Good
**Technical Quality:** Very Good
**Clarity Of Presentation:** Very Good
**Impact:** 3

**Recommendation:**

Weak Accept: I recommend accepting the paper, but will not argue for my recommendation if the majority of other reviewers have a different opinion.

**Review:**

Strengths:

- Use FCN to capture granular dynamics, and enable a fixed-resolution approach for efficiency.
- Use curvilinear trajectory optimization for obstacle avoidance during manipulation.

Weaknesses:

- Fixed resolution FCN may lose some details for smaller granular.

I have a question about the density field representation, does there exist a trade-off between field resolution and dynamic modeling accuracy? Can FCN model small particles well?

**Quality Of The Limitations Section:**

Limitations are addressed clearly

**Questions For Rebuttal:**

I am curious about the limitation of CNN resolution for the manipulation tasks, e.g. is it possible for this framework to capture dynamics of relatively smaller granulars?

**Robotics Focus:**

Sufficient demonstration on hardware

**Summary Of Paper:**

The paper proposes a method that uses a Fully Convolutional Network (FCN) to encode granular dynamics, also it utilizes differential render to construct loss for dynamics and trajectory optimization. This method is efficient compared with particle-based methods and it demonstrates good performance for simulation and real-world tasks.



**Summary Of Recommendation:**

I think this is an interesting paper that can be accepted. Overall the experiments show the effectiveness of the method and the framework is sound.

---

### Official Review · Reviewer_khMs · 2023-07-18

**Confidence:** 3
**Originality:** Good
**Technical Quality:** Good
**Clarity Of Presentation:** Good
**Impact:** 3

**Recommendation:**

Weak Accept: I recommend accepting the paper, but will not argue for my recommendation if the majority of other reviewers have a different opinion.

**Review:**

Strength

The authors make a real-world implementation and the results shown in the supplementary video look impressive.

This pipeline is end-to-end differentiable so can utilize the gradient-based optimization for planning, which is efficient and powerful.

In the experiments, the proposed method has decent performance and outperforms other baselines. It can also achieve reasonable results in versatile environments.

Weakness

The dynamics networks lack interpretability. The correctness of the learned state transition isThe real-world experiment is great. I’m curious how much effort is used in sim-to-real transferring. Moreover, what is the success rate? Can the robot arm consistently achieve the goals? hard to verify.


**Quality Of The Limitations Section:**

Limitations are addressed clearly

**Questions For Rebuttal:**

The real-world experiment is great. I’m curious how much effort is used in sim-to-real transferring. Moreover, what is the success rate? Can the robot arm consistently achieve its goals?

**Robotics Focus:**

Sufficient demonstration on hardware

**Summary Of Paper:**

This paper proposes a learning-based method for granular material manipulation. The scene is represented by voxels, and a fully convolutional neural network operates on a density-filed-based representation for the object piles. The dynamics network learns the state transition and another differentiable rendering module can map the state into the trajectory. The entire pipeline can be used to perform trajectory optimization.

**Summary Of Recommendation:**

The idea of this paper is clear and intuitive. It is nice to learn a fully differentiable simulation and rendering pipeline and further use the learned model to perform gradient-based optimization. The experiments are convincing that the methods can be implemented on real robot arms, stay robust under different settings, and outperform baseline methods. Therefore, I recommend acceptance.

---

### Official Review · Reviewer_LnEd · 2023-07-20

**Confidence:** 3
**Originality:** Good
**Technical Quality:** Good
**Clarity Of Presentation:** Very Good
**Impact:** 3

**Recommendation:**

Weak Accept: I recommend accepting the paper, but will not argue for my recommendation if the majority of other reviewers have a different opinion.

**Review:**

Strengths:
- Main contribution is simple and fits to the task at hand.
- Experiments are thorough, comparing to several state of the art methods and demonstrating clear performance gains.

Weaknesses:
- Setup assumes full observability of granular material and obstacles.
- Analysis is missing some clarity in explanation.

The FCN contribution builds on well understood principles and applies it effectively to the granular object manipulation task, and in a novel way. One form of related work that should be added is Reinforcement Learning based methods (e.g., "Learning to Manipulate Object Collections Using Grounded State Representations," Matthew Wilson and Tucker Hermans, CoRL, 2020).

Another work to be aware of is "Dynamic-Resolution Model Learning for Object Pile Manipulation" by Yixuan Wang, Yunzhu Li, Katherine Driggs-Campbell, Li Fei-Fei, and Jiajun Wu, RSS 2023, which proposes using predicted resolutions of particles to avoid inefficiency in GNN-based dynamics and demonstrated impressive real world results. If possible, this work should be compared to, as this is (to the reviewer's knowledge) is the current SOTA in GNN-based granular dynamics.

The theoretical analysis performed in Sec. 3.2 is difficult to follow. In particular, it is not clear how to related Eqs. 1 and 2 and the derivation of the correlation term following Eq. 2 is also unclear. Further explanation of why these terms are important along with their derivations (potentially in appendix) would strengthen quality of result.

The experiments demonstrate significant results, including generalization to new environments, good performance scaling, and importantly the ability to reason over obstacles and multiple task objectives with a single method. The performance is encouraging for the use of learned dynamics trajectory optimization in granular material manipulation.

Other notes:
- Sec 3.2 line 137 says Connected instead of Convolutional, which is important to be clear about.
- Appendix B.1, line 441 I believe should refer to Fig. 3.

**Quality Of The Limitations Section:**

Limitations are addressed clearly

**Questions For Rebuttal:**

- How does this method compare performance wise to the method proposed by "Dynamic-Resolution Model Learning for Object Pile Manipulation" by Yixuan Wang, Yunzhu Li, Katherine Driggs-Campbell, Li Fei-Fei, and Jiajun Wu, RSS 2023.
- Please provide clarifications on the correlation analysis.
- In Fig. 4, why does sequential prediction outperform single - wouldn't multiple predictions be more difficult?
- For the obstacle avoidance task, how does Opt based planning (as opposed to curve) perform on this task? Given the results in Table 1, is it potentially easier to optimize/more accurate to model straight pushes?
- In Table 4, what was the success rate?
- How are orientations determined for the B-spline curve method? Appendix A seems to highlight just position interpolation.

**Robotics Focus:**

Sufficient demonstration on hardware

**Summary Of Paper:**

This work proposes a Fully Convolutional Network (FCN) for learning dynamics of granular object piles. The key intuition is that, unlike alternative modeling choices (fully connected, graph, etc.), the inductive biases of FCN match the local nature of the granular dynamics. States and actions are rendered into grids and the resulting dynamics are utilized for curvilinear trajectory optimization. The work provides theoretical and experimental results validating that this representation is well fitted to the problem and outperforms alternative methods at granular object sweeping, including with obstacles.

**Summary Of Recommendation:**

The proposed solutions is simple and effective and the experiments are thorough and show strong performance compared to baselines. I am unsure of the technical soundness of some of the theoretical analysis surrounding the inductive biases and have asked for clarification; I would want to check that more thoroughly before accepting. Additionally, I have asked for comparison to a recently released work from RSS 2023 that updates the SOTA in GNN-based dynamics which would provide a more up to date baseline for the method. Assuming good performance there, I think the paper presents a strong case.

---

### Decision · Program_Chairs · 2023-08-30

**Decision:**

Accept (Poster)

**Comment:**

This paper learns a dynamics model for granular materials, assuming a top-view camera observation. The model proposed in the paper is a Fully Convolutional Network, which seems to perform well, but relies heavily on the top-view assumption so that the pile of granular material can be treated as a 2.5D heightfield. Manipulation of the pile is done in a myopic way by optimizing the parameters of a curvilinear trajectory, which leads to inefficient motion, as evidenced by the accompanying videos. Overall, I think the main contribution of this paper is advocating for the FCN as a structural design choice for modeling granular media. I recommend that the paper be accepted as a poster.